# A New Biocontrol Agent *Bacillus velezensis* SF334 against Rubber Tree Fungal Leaf Anthracnose and Its Genome Analysis of Versatile Plant Probiotic Traits

**DOI:** 10.3390/jof10020158

**Published:** 2024-02-17

**Authors:** Muyuan Wang, Yikun Zhang, Haibin Cai, Xinyang Zhao, Zhongfeng Zhu, Yichao Yan, Ke Yin, Guanyun Cheng, Yinsheng Li, Gongyou Chen, Lifang Zou, Min Tu

**Affiliations:** 1Shanghai Collaborative Innovation Center of Agri-Seeds, School of Agriculture and Biology, Shanghai Jiao Tong University, Shanghai 200240, China; wmy040549@sjtu.edu.cn (M.W.); zyk982833375@sjtu.edu.cn (Y.Z.); zhuzhongfeng@sjtu.edu.cn (Z.Z.); yichao.yan@sjtu.edu.cn (Y.Y.); yin.ke@sjtu.edu.cn (K.Y.); guanyuncheng-yjw@sjtu.edu.cn (G.C.); yinshengli@sjtu.edu.cn (Y.L.); gyouchen@sjtu.edu.cn (G.C.); 2National Key Laboratory for Tropical Crop Breeding, Rubber Research Institute, Chinese Academy of Tropical Agricultural Sciences, Sanya 572024, China; haibin_cai@163.com; 3School of Agriculture, Yangtze University, Jingzhou 434000, China; xyzhao0329@126.com; 4State Key Laboratory of Microbial Metabolism, Shanghai Jiao Tong University, Shanghai 200240, China; 5Sanya Research Institute, Chinese Academy of Tropical Agricultural Sciences, Sanya 572020, China

**Keywords:** *Bacillus velezensis*, anthracnose, rubber tree, *Colletotrichum siamense*, *Colletotrichum australisinense*

## Abstract

Natural rubber is an important national strategic and industrial raw material. The leaf anthracnose of rubber trees caused by the *Colletotrichum* species is one of the important factors restricting the yields of natural rubber. In this study, we isolated and identified strain *Bacillus velezensis* SF334, which exhibited significant antagonistic activity against both *C*. *australisinense* and *C*. *siamense*, the dominant species of *Colletotrichum* causing rubber tree leaf anthracnose in the Hainan province of China, from a pool of 223 bacterial strains. The cell suspensions of SF334 had a significant prevention effect for the leaf anthracnose of rubber trees, with an efficacy of 79.67% against *C*. *siamense* and 71.8% against *C*. *australisinense*. We demonstrated that SF334 can lead to the lysis of *C*. *australisinense* and *C*. *siamense* mycelia by causing mycelial expansion, resulting in mycelial rupture and subsequent death. *B*. *velezensis* SF334 also harbors some plant probiotic traits, such as secreting siderophore, protease, cellulase, pectinase, and the auxin of indole-3-acetic acid (IAA), and it has broad-spectrum antifungal activity against some important plant pathogenic fungi. The genome combined with comparative genomic analyses indicated that SF334 possesses most genes of the central metabolic and gene clusters of secondary metabolites in *B*. *velezensis* strains. To our knowledge, this is the first time a *Bacillus velezensis* strain has been reported as a promising biocontrol agent against the leaf anthracnose of rubber trees caused by *C*. *siamense* and *C*. *australisinense*. The results suggest that *B*. *velezensis* could be a potential candidate agent for the leaf anthracnose of rubber trees.

## 1. Introduction

The rubber tree (*Hevea brasiliensis*) is a perennial tropical tree that is the primary source of natural rubber, an important industrial raw material [1]. The anthracnose of rubber trees caused by the genus *Colletotrichum* is one of the main diseases restricting rubber output [2]. The disease mostly affects the leaves, stems, and fruits, causing leaf shedding, stem spotting, branch drying, and fruit rot and eventually resulting in reduced yields of rubber trees [3]. Currently, the disease has been reported to occur in several countries, including Sri Lanka, China [4], India [5], and Brazil [6], and *C. gloeosporioides* and *C. acutatum* were considered to be the main causative agents [7]. In recent years, researchers have gradually discovered and identified some new and dominant species. For instance, *C. acutatum* and *C. gloeosporioides* causing anthracnose in rubber trees were recorded for the first time in India [8]. Four novel pathogenic species of *C. laticiphilum*, *C. nymphaeae*, *C. citri*, and *C. simmondsii*, all belonging to the *C. acutatum* complex, were found in Sri Lanka [9]. *C. siamense* [10], *C. australisinense* [10], *C. wanningense* [7], and *C. cliviae* [11], which can cause leaf anthracnose in rubber trees, were found in China. *C. siamense* and *C. australisinense* are thought to be the main causal species of anthracnose in Hainan province, which is one of the main rubber tree-growing areas in China [10]. The classification of species complexes associated with the anthracnose of rubber trees is complicated, and dominant species in different regions have great differences [3], which introduces certain difficulties to disease control; therefore, the development of effective control measures has become a key priority.

The control of anthracnose-causing rubber trees mainly focuses on chemical, agricultural, and biological controls and breeding for disease resistance. Generally, chemical controls are effective, but they also cause some problems such as environmental pollution and the emergence of drug resistance. Disease-resistant breeding is economical and effective but requires a long research and development process [12]. Agricultural strategies focus on land management and environmental improvement [6], which are economical but slow in effect and have geographical and seasonal characteristics. In contrast, biological controls have brighter application prospects due to the advantages of strong selectivity and environmental friendliness [13]. Some antagonistic strains derived from bacterial and fungal sources were effective in controlling the anthracnose of rubber trees. Three strains of *Streptomyces* YQ-33, QZ-9, and WZ-2 with high antagonistic activity against *C. siamense* were isolated and identified [14]. The lipopeptide produced by *Bacillus subtilis* Czk1 could significantly control the anthracnose of rubber trees [15], and it was more effective when combined with “Rootcon” [16], a common chemical antiseptic. An endophytic fungal strain *Epicoccum dendrobii* SMEL1 from a young healthy branch of Chinese fir (*Cunninghamia lanceolata*) was found to have a highly antagonistic effect on *C. gloeosporioides* [17]. Therefore, the identification of novel antagonistic strains against species of *Colletotrichum* will provide more microbial resources for the biological control of anthracnose-causing rubber trees.

This study aims to identify the biocontrol strain that has an effective control effect on the leaf anthracnose of rubber trees in the Hainan province of China. In this study, we obtained a *Bacillus velezensis* strain SF334 that exhibited significant antagonistic activity against both *C*. *australisinense* and *C*. *siamense*, two major pathogens causing leaf anthracnose of rubber trees in the Hainan province of China, from 69 candidate strains from a pool of 223 bacterial strains using the paper filtering method. SF334 exhibited a great biocontrol potential in the prevention of leaf anthracnose in rubber trees. We explored the mechanism of SF334 inhibiting the mycelial growth of *C. australisinense* and *C. siamense* and analyzed the plant probiotic characterizations and antagonistic spectrum of SF334. In addition, we completed whole genome sequencing and conducted the comparative genomic analysis of SF334 with its closely related *B*. *velezensis* strains. It was indicated that SF334 has genes associated with antimicrobial and plant growth promotion. These works provided a new microbial resource for the biological control of the leaf anthracnose of rubber trees and laid a preliminary foundation for subsequent applications in forests and fields.

## 2. Materials and Methods

### 2.1. Strains and Growth Conditions

All bacterial strains were cultured in nutrient agar (NA, 5 g polypeptone, 10 g sucrose, 1 g yeast extract, 3 g beef extract, 15 g agar, 1000 mL distilled water, pH: 7.0), nutrient broth (NB, NA without agar), or Lysogeny broth (LB) medium at 28 °C. The fungal pathogens causing the leaf anthracnose of rubber trees, *C*. *siamense* CS-DZ-1 and *C*. *australisinense* CA-DZ-5, were isolated from experimental fields at the National Rubber Germplasm Repository in Hainan, and they were grown on potato dextrose agar (PDA) or in PDB (PDA without agar) medium at 25 °C. Other fungal strains, including *Magnaporthe oryzae* causing rice blast, *Fusarium oxysporum* f. sp. *Spcucumerinum* causing root rot disease of cucumber, *F*. *graminearum* causing fusarium head blight, *Alternaria solani* causing early blight of potato, *Phytophthora capsici* causing pepper phytophthora blight, and *Botrytis cinerea* causing gray mold disease of vegetables, were cultured on PDA medium at 25 °C.

### 2.2. Screening and Identification of the SF334 Strain

A library that includes 223 bacterial isolates has been established in our previous work [18]. *C*. *siamense* CS-DZ-1 and *C*. *australisinense* CA-DZ-5 were used to screen antagonistic strains. The fungal piece was placed in the center of the PDA media, and the filter papers were placed 2 cm away from the center. In total, 5 μL of the bacterial solution (OD_600_ = 2.0) was added to the filter papers located in the left and right direction, and the same volume of NB medium was added to the top and bottom as a negative control. Three replicates in each group were cultured at 28 °C for 5–7 days. The inhibition rates were calculated according to the growth diameters, and SF334 with strong antagonistic activity against both *C. siamense* and *C. australisinense* was screened. SF334 was isolated from the orchard soil sample collected from Haidian Harbor Garden in Haikou City of Hainan province, China, on 8 November 2018.

The *16S rRNA* sequence of SF334 was amplified using the universal primers of 27F and 1492R according to our previous protocol [19]. The sequence was used for searches in the National Center for Biotechnology Information (NCBI) database, and the sequences of the *16s rRNA* of 39 strains with the lowest E-value were selected as the reference sequences. The results were finally imported into MEGA 11 software to construct a phylogenetic tree using the NJ (1200 bootstrap) method. Genome-wide phylogenetic trees were constructed using the TYGS platform (https://tygs.dsmz.de/, accessed on 15 May 2023). Average nucleotide identity (ANIb) and DNA-DNA hybridization (DDH) analyses were performed through online websites http://jspecies.ribohost.com/jspeciesws/, accessed on 15 May 2023 and https://ggdc.dsmz.de/, accessed on 15 May 2023, respectively.

### 2.3. Genomic Sequencing, Assembly, and Annotation of SF334

A single colony of SF334 was inoculated in NB medium and incubated overnight at 28 °C, 200 rpm min^−1^. The resulting bacterial solution was transferred to a new NB medium at a ratio of 1:100 for further incubation until the liquid reached the logarithmic stage (OD_600_ is between 0.4 and 0.8). The bacterial solution of SF334 was centrifuged at 4 °C, 6000 rpm min^−1^ for 10 min and collected, then washed with 1 × PBS buffer twice, and finally removed the supernatant. The resulting bacterial precipitates were frozen in liquid nitrogen for 15 min and then sent to the Beijing Genomics Institute for whole genome sequencing through the PacBio Sequel II platform 4.

After obtaining raw data, low-quality sequences, joint sequences, etc., were removed. Genome assembly was performed by Personalbio (Shanghai, China). Open reading frames were predicted using GeneMark S (http://exon.gatech.edu/GeneMark/, accessed on 1 May 2023); non-coding RNAs were predicted using tRNAscan-SE v1.3.3, Barrnap, and Rfam databases; tandem repeats were predicted using the Tandem Repeats Finder. The assembled sequences were annotated in GO (Gene Ontology), KEGG (Kyoto Encyclopedia of Genes and Genomes), COG (Clusters of Orthologous Groups), Swiss-Prot, NR (Non-Redundant Protein Database), TCDB (Transporter Classification Database), CAZy (Carbohydrate-Active enZYmes Database), CARD (The Comprehensive Antibiotic Resistance Database), antiSMASH, and signal P6.0 databases for functional gene analysis and annotation.

The complete genome sequence of *B. velezensis* SF334 was deposited in GenBank under accession number CP125289.

### 2.4. Comparative Genomic Analysis

The genomic features of SF334 and the model strains *B. velezensis* FZB42, *B. velezensis* SQR9, *B. amyloliquefaciens* DSM7, and *B. subtilis* 168 were compared using the GeneMark (http://exon.gatech.edu/GeneMark/, accessed on 25 May 2023) and RAST (https://rast.nmpdr.org/, accessed on 25 May 2023) databases, and genomics covariance analysis was performed using the Mauve (https://darlinglab.org/mauve/mauve.html, accessed on 25 May 2023) database. Pan-genomic analysis was performed using the BPGA (v.1.3) (https://iicb.res.in/bpga/index.html, accessed on 25 May 2023) tool. The comparative analysis of carbohydrate-active enzymes was carried out using dbCAN2, an online annotation tool available at http://bcb.unl.edu/dbCAN2/, accessed on 25 May 2023, and secondary metabolite gene cluster synthesis was conducted using the online antiSMASH software (https://antismash.secondarymetabolites.org/, accessed on 20 October 2023).

### 2.5. Biocontrol Assays

For in vitro biocontrol assays, a total of 36 leaves from 1-m-high Brazilian rubber trees that were obtained from the sprout of the “RRIM600” cultivar were inoculated with an agar disk containing the mycelium of *C*. *siamense* and *C*. *australisinense*. Six agar disks were inoculated on each leaf and cultured at 28 °C for 72 h. The diameters of the lesions were measured. Similar experiments were conducted on live Brazilian rubber trees using the 10^6^ conidium mL^−1^ suspension of *C*. *siamense* and *C*. *australisinense*. The SF334 treatment (Tre) strategy meant that leaves were sprayed with the cell suspensions (CSs) of SF334 (OD_600_ = 1.0) 24 h after inoculation with *C*. *siamense* and *C*. *australisinense*. The SF334 preventive (Pre) strategy indicated that leaves were sprayed with the CSs of SF334 24 h before inoculation with *C*. *siamense* and *C*. *australisinense*. The inhibitory percentages (IPs) were calculated using the following formula: IP = (1 − diameters of treatment/diameters of control) × 100%. The IP was calculated using 36 technical replicates per assay.

### 2.6. Hyphal Digestion Observations

Three bacterial solutions, with each measuring 50 μL of SF334 with different concentrations (OD_600_ = 1.0, 2.0, and 4.0), were dropped on the PDA medium covered with the mycelium of *C. siamense* and *C. australisinense* at a distance of about two centimeters away from the center, and the same volume of NB medium was added to the top as a negative control. The degradation condition of the mycelium was photographed every 30 min.

### 2.7. Microscopic Observations

The fungal pieces of *C. siamense* and *C. australisinense* were added to PDB media, and they were cultured at 28 °C and 180 rpm/min for 2 days; then, they were mixed with the bacterial solution of SF334 (OD_600_ = 1.5) for further incubation. The samples from the mixture at 1 h postincubation (hpi), 3 hpi, and 6 hpi were stained with 0.05% Evans blue for 2 h; then, they were observed under an optical microscope. The same experiments were conducted as above using the cell-free supernatants (CFSs) of SF334 prepared from the bacterial solution of OD_600_ = 1.5.

The samples at 1 hpi and 6 hpi were chosen for observation via a scanning electron microscope (SEM). First, the samples were treated overnight with 2.5% glutaraldehyde, and they were fixed by 1% osmium acid. Then, they were dehydrated using 30% ethanol, 50% ethanol, 70% ethanol, 90% ethanol, and anhydrous ethanol in turn. Finally, they were dried using a CO_2_ critical point dryer. The samples were observed using SEM (NOVA NanoSEM 230) at the Instrumental Analysis Center, Shanghai Jiao Tong University.

### 2.8. Analysis of Plant Probiotic Characteristics

The cellulase assay medium (20 g carboxymethyl cellulose, 2 g K_2_HPO_4_, 0.5 g KH_2_PO_4_, 2 g (NH_4_)_2_SO_4_, 6 g NaCl, 0.1 g CaCl_2_, 0.1 g MgSO_4_·7H_2_O, 20 g agar, 1000 mL distilled water, pH 7.0-7.5); chitinase assay medium (15 g chitin colloid, 1.36 g KH_2_PO_4_, 1 g (NH_4_)_2_SO_4_, 0.3 g MgSO_4_·5H_2_O, 3 g yeast extract, 15 g agar, 1000 mL distilled water, pH 7.0); protease assay medium (A solution: 10% defatted milk, B solution: 3% agar solution; C solution: 0.2 M phosphate buffer, pH = 7.0, A, B, and C were fully mixed); PKO medium (0.5 g (NH_4_)_2_SO_4_, 0.2 g KC1, 0.2 g NaC1, 4 g Ca_2_(PO_4_)_3_, 0.1 g MgSO_4_·7H_2_O, 0.0004 g MnSO_4_, 0.0002 g F eSO_4_, 0.5 g yeast extract, 10 g sucrose, 15 g agar, 1000 mL distilled water, pH: 7.0); potassium-solubilizing medium (1 g potassium feldspar powder, 1 g CaCO_3_, 2 g Na_2_HPO_4_, 1 g (NH_4_)_2_SO_4_, 0.5 g MgSO_4_·7H_2_O:, 10 g sucrose, 0.5 g yeast extract, 15 g agar, 1000 mL distilled water, pH: 7.0); ferritin secretion capacity assay medium (3 g casein acids hydrolysate: 1 mL 1 mM CaCl_2_, 20 mL 1 mM MgSO_4_, 50 mL CAS A solution (1 mM CAS, 4 mM HDTMA, 0.1 mM FeCl_3_); and 5 mL CAS B solution (0.1 mM phosphate buffer, pH = 7.0), 2 g sucrose, 20 g agar, 1000 mL distilled water) were prepared for the analyses of cellulase activity, pectinase activity, protease activity, phosphorus solubilizing activity, potassium solubilizing activity, and siderophore production, respectively. A 50 μL bacterial solution of SF334 (OD_600_ = 2.0) was filled into an Oxford cup located in the middle of the plate, and the plate was incubated at 28 °C for 48 h to observe whether the hydrolysis circle was produced. If a hydrolytic halo is generated, it indicates that the corresponding enzymatic reaction has occurred, and SF334 has the corresponding enzymatic activity.

For IAA analysis, the bacterial solution of SF334 was directly inoculated into the YM medium (5 g mannitol, 0.05 g NaCl, 0.25 g K_2_HPO_4_, 1.5 g yeast extract, 0.05 g L-tryptophan, pH 7.0, 1000 mL distilled water) and then incubated at 28 °C and 135 rpm min^−1^ for 96 h. The supernatant was mixed with the colorimetric solution (1.5 mL 0.5 M FeCl_3_, 30 mL H_2_SO_4_, 50 mL distilled water) in a 1:1 volume ratio to observe the color changes. If the solution appears pink, the solution contains indolic compounds. The IAA concentration was calculated according to the standard curve formula y = 0.0157x + 0.1137, where Y is the measured UV absorbance in OD_530_ and X is the IAA concentration value.

### 2.9. Antifungal Activity Assays

*M*. *oryzae* causing rice blast, *F*. *oxysporum* f. sp. *spcucumerinum* causing root rot disease of cucumber, *F*. *graminearum* causing fusarium head blight, *A*. *solani* causing early blight of potato, *P*. *capsici* causing pepper phytophthora blight, and *B*. *cinerea* causing gray mold disease of vegetables were used as the target pathogens. The antifungal activity of SF334 was measured using our previous protocol [20].

## 3. Results

### 3.1. Screening and Identification of Strain SF334 That Exhibits Highly Antagonistic Activity against C. siamense and C. australisinense

To screen the antagonistic strains of *C. siamense* and *C. australisinense*, which are major pathogens causing the leaf anthracnose of rubber trees in the Hainan province of China, we obtained 69 candidate strains from a pool of 223 bacterial strains using the filtering paper method. Among these strains, we found a strain designated as SF334 that exhibited significant antagonistic activity against both *C*. *siamense* strain CS-DZ-1 and *C*. *australisinense* stain CA-DZ-5 with an average inhibition rate of 66.17% and 68.15%, respectively (Figure 1A and Appendix A).

To clarify the taxonomic status of SF334, we carried out the sequence alignment analysis of the *16S rRNA* gene and constructed the corresponding phylogenetic tree (1200 bootstrap). The results indicated 99.94% homology between SF334 and *Bacillus velezensis* FZB42, a model strain of *B. velezensis* [21], and the phylogenetic tree showed that SF334 and *B. velezensis* FZB42 were in the same branch (Appendix A), tentatively confirming that SF334 is *B. velezensis*. We further sequenced the whole genome of SF334 (Figure 1B) and subsequently selected 13 *Bacillus* strains, including 5 strains of *B. velezensis* for ANIb and DDH analyses. The ANIb values of SF334 with 5 strains of *B. velezensis* exceeded 95%, the threshold for species demarcation, and the DDH values also exceeded the accepted species threshold of 70% (Figure 1C). The ANIb and DDH values did not go above the critical classification value for species when SF334 is involved compared to the other strains (Figure 1C). These results further confirmed that SF334 belongs to *B. velezensis*.

### 3.2. Assessment of SF334 as Effective Biocontrol Agent for Leaf Anthracnose of Rubber Tree Caused by C. siamense and C. australisinense

We used SF344 to perform biocontrol tests on Brazilian rubber trees. In the experiments based on detached leaves, we sprayed the cell suspensions (CSs) of SF334 on the in vitro leaves of rubber trees and implemented prevention (Pre) and treatment (Tre) strategies. Compared with the control group (CK), the prevention efficacy of the CSs of SF334 for leaf anthracnose caused by *C*. *siamense* was 60.08%, and the control efficacy was 39.74% (Figure 2A,C). The prevention and control effect of the CSs of SF334 for leaf anthracnose caused by *C. australisinense* was 72.15% and 40.6% (Figure 2B,D), respectively.

Further, we conducted similar experiments on live Brazilian rubber trees and found that the prevention efficacies of the CSs of SF334 for anthracnose caused by *C*. *siamense* and *C. australisinense* were 79.60% and 71.96% (Figure 3A–D), whereas the control efficacies were 39.60% and 40.54% (Figure 3A–D), respectively, which was consistent with the results in vitro. These results suggested that SF334 is an effective biocontrol agent for the protection of rubber trees against leaf anthracnose.

### 3.3. B. velezensis SF334 Inhibits C. siamense and C. australisinense by Disrupting Growth of Mycelium

To further explore the antagonistic mechanism of SF334 against *C*. *siamense* and *C*. *australisinense*, we attempted to drop the bacterial solution of SF334 into the growing hyphae of *C*. *siamense* and *C*. *australisinense* on PDA media. The hyphae of both *C*. *siamense* and *C*. *australisinense* were gradually degraded with the extension of interaction time (Appendix A) and were completely disrupted at about 3 h postinteraction (hpi) (Figure 4A). The rate of the hyphal degradation of *C. siamense* and *C. australisinense* was generally accelerated with an increase in bacterial concentration; however, the difference was weak when the OD_600_ of the bacterial concentration of SF334 was either 2.0 or 4.0 (Figure 4A and Appendix A), suggesting that the ability of SF334 to cause hyphal degradation may be close to its peak.

The observation via optical microscopy showed that SF334 caused the mycelial expansion of *C. siamense* and *C. australisinense* (mainly in the apical part and a few in the middle) (Appendix A). Evans blue staining showed that the expanded mycelia were stained blue after interaction with SF334, indicating the death of mycelia (Figure 4B). With the prolongation of interaction time, mycelial expansion became more obvious, and the proportion of mycelial death gradually increased (Figure 4B and Appendix A).

Since the mycelia of pathogenic *C. siamense* and *C. australisinense* depend on the apex’s growth, the apical expansions severely restrict the growth of mycelia. To further investigate the phenotype of mycelial death, we used scanning electron microscopy to observe the interactions of SF334 with *C. siamense* and *C. australisinense*. The observations showed that the mycelia of *C. siamense* and *C. australisinense* were regular in shape and full; with normal spore germination, the tip of a single mycelium had a prominent growing point and could grow new mycelium continuously (Figure 5). However, the tips of the mycelia of *C. siamense* and *C. australisinense* were deformed and bulbous when interacting with SF334 for 1 h or 6 h (Figure 5). The cell walls of the mycelia shrank, and the contents of cells leaked obviously at 6 hpi (Figure 5). Similar mycelial expansion phenotypes were observed when we used the cell-free supernatants (CFSs) of SF334 to interact with *C. siamense* (Figure 6A), and when we sprayed the CFSs of SF334 on the live leaves of Brazilian rubber trees (Figure 6B). From these results, we speculated that the extracellular active compounds secreted by SF344 may cause deformations of the mycelium, resulting in mycelial rupture and subsequent death.

### 3.4. Analysis of Plant Probiotic Characterizations and Antagonistic Spectrum of B. velezensis SF334

To further examine the potential of SF334 for biocontrol applications, we conducted analyses of plant probiotic characterizations and the antagonistic spectrum against fungal pathogens. The analyses of some plant probiotic traits showed that SF334 could secrete siderophore and protease (Figure 7A); however, they cannot degrade inorganic phosphorus and potassium (data not shown). SF334 also could produce the auxin of indole-3-acetic acid (IAA) (Figure 7B), which is capable of stimulating plant growth. Our quantitative determination showed that SF334 could produce IAA with 9.45 mg/L. SF334 also could secrete cellulase and pectinase (Figure 7A), indicating that SF334 can degrade the components of cell walls from fungi and oomycetes. Whether the extracellular pectinase is responsible for the observed phenotypes of mycelial digestion remains to be determined. Cellulases can also break plant tissues, which might be beneficial if “good” bacteria are allowed to enter into the plant. The antagonistic experiments based on the filter paper method showed that the inhibition rate of SF334 was 59.63% against *Magnaporthe oryzae* causing rice blast, 50.93% against *Fusarium oxysporum* f. sp. *spcucumerinum* causing the root rot disease of cucumber, 56.67% against *F*. *graminearum* causing fusarium head blight, 59.26% against *Alternaria solani* causing the early blight of potato, 51.48% against *Phytophthora capsici* causing pepper phytophthora blight, and 61.85% against *Botrytis cinerea* causing the gray mold disease of vegetables (Figure 7C). These results indicated that *B*. *velezensis* SF334 has broad-spectrum antifungal activity and is a versatile plant probiotic bacterium.

### 3.5. Genomic Features and Functional Gene Analysis of B. velezensis SF334

The genome of SF334 consists of a 4,078,641 bp circular chromosome without plasmids (Figure 1B), with a GC content of 46.5%, and 4,164 protein-coding sequences (CDSs), occupying 88.76% of the total chromosome length. SF334 has 86 tRNA genes, 33 sRNA genes, 9 genes for 5S rRNA, 9 genes for 16S rRNA, and 9 genes for 23S rRNA (Table 1).

The COG annotation results revealed that there are 3022 genes annotated relative to the COG database in SF334 (Table 1), accounting for 72.57% of the number of predicted genes. These genes were annotated relative to 24 COG entries, of which 18 entries had more than 100 annotated genes (Appendix A). The most abundant genes are involved in amino acid transport and metabolism, followed by major functional and transcriptional carbohydrate transport and metabolic processes. SF334 has genes encoding peptidoglycan/xylan/gibberellin deacetylase (COG0726), β-glucanase (COG2273), and β-mannanase (COG4124), which may be associated with the hydrolysis of the fungal cell wall. Besides this, SF334 has two iron carrier transport systems (COG0609 and COG1120) that facilitate competition for iron ions from the environment to achieve competitive antibacterial purposes [22].

The GO annotation results showed that 2376 genes were annotated to the GO database in SF334 (Table 1), accounting for 57.06% of the number of predicted genes. Among them, 1297 genes are involved in cellular processes, followed by metabolic processes, catalytic activity, and binding processes (Appendix A). SF334 has gene functions associated with fungal cell structure hydrolase activity, such as carbohydrate metabolism (GO:0005975), a chitin-binding domain (GO:0008061), and a protease core complex (GO:0005839), predicting that SF334 may inhibit pathogenic fungi through antagonistic effects.

The KEGG analysis showed that SF334 has 2554 genes annotated relative to 42 pathways, accounting for 61.33% of the total number of genes (Appendix A), indicating that SF334 has abundant substance metabolic pathways and can use a variety of substances to meet its own needs, making it well adapted to the environment. In addition, SF334 also has genes associated with growth hormone synthesis, such as *trpA*, *trpB*, *trpC*, and *aldh* (Appendix A), and it is assumed that SF334 may use tryptophan as a precursor substance to synthesize indole-3-acetic acid through the indole-3-pyruvate pathway (IPA pathway).

CAZy analysis predicted that SF334 has 200 genes encoding carbohydrate-active enzymes, occupying 2.47% of the total number of genes and containing 59 genes related to carbohydrate synthesis and 141 genes related to hydrolysis (Appendix A). Moreover, the SF334 genome contains 5 cellulose biohydrolases (EC 3.2.1.91), 9 tributylases (EC 3.2.1.14), 6 endoglucanases (EC 3.2.1.4), 15 lysozymes (EC 3.2.1.17), 3 mannanases (EC 3.2.1.78), and other related genes that may enable SF334 to have a strong ability to degrade the fungal cell wall. In addition, SF334 contains genes related to alglucan synthesis (EC 3.2.1.28), which are closely related to the stress resistance of the strain [23].

In summary, these analyses suggest that SF334 may be a multifunctional plant probiotic strain with probiotic and biocontrol properties.

The antiSMASH analysis revealed that a total of 16 gene clusters related to secondary metabolite synthesis were obtained from SF334 (Figure 8 and Table 2), among which 9 known secondary metabolite gene clusters (macrolactin H, bacillaene, bacillomycin-D, fengycin, difficidin, bacillothiazole A, bacillibactin, amylocyclicin, and bacilysin) showed 100% similarity to known secondary metabolite synthesis gene clusters included in the database and another other 2 (surfactin and plantazolicin) reached more than 80% similarity, indicating that SF334 has a high probability of synthesizing these secondary metabolites. Surfactin [24], fengycin [25], difficidin [26], bacilysin [27], and bacillibactin [28] all possess antibacterial and antifungal activities, with antibacterial mechanisms involving direct inhibition, lysis, and competitive effects.

### 3.6. Comparative Genomic Analysis of B. velezensis SF334 with Other Representative Bacillus Strains

To investigate the differences between SF334 and other genetically related *Bacillus* species, we selected four currently widely studied bacteria, *B. velezensis* FZB42, *B. velezensis* SQR9, *B. amyloliquefaciens* DSM7, and *B. subtilis* 168, for comparative analysis with SF334. The genomic characteristics of SF334 and the reference strains showed that the genome size of SF334 is between *B. velezensis* FZB42 and *B. velezensis* SQR9 (Appendix A). The genomic GC content, coding region density, tRNA, rRNA, and the number of repeat regions of SF334 are similar to those of FZB42 (Appendix A). Further collinearity analysis of SF334, FZB42, and SQR9 showed that the genomes of SF334 are highly similar to those of SQR9 and FZB42, with direct linear correspondence for most genes and gene rearrangements, such as translocations in a few cases (Figure 9A). The genomic similarity between SF334 and SQR9 is higher, indicating closer relativity.

Pan-genomic analysis using BPGA showed that SF334 has 2807 core genes and 352 endemic genes between SF334 and the four reference genomes, with high similarity among the five genomes (Figure 9B). COG annotation results showed that the core genes are mainly distributed in the general functional cluster (R), amino acid transport and metabolism (E), and transcription (K) functional classifications, with the lowest proportion in cell cycle, division, and chromosome assignment (D); endemic genes are mainly distributed in general functional clusters (R), transcription (K), and carbohydrate transport and metabolism (G) (Appendix A). The results of KEGG annotation revealed that the core genes are mainly annotated relative to carbohydrate metabolism and amino acid metabolism pathways, and only core genes are found in transcriptional and environmental adaptation pathways; the unique genes are mainly annotated relative to carbohydrate metabolism and cell membrane transport pathways (Appendix A). Combining the annotation results of COG and KEGG, the five strains have similar amino acid transport metabolic pathways, transcriptional pathways, and environmental adaptation pathways, which indicates functional similarity among strains. Each strain has its specific carbohydrate metabolic pathway and cell membrane transport pathway, suggesting that SF334 may have different functional properties, such as unique carbohydrate-active enzymes.

We conducted the comparative analysis between SF334 and the other four *Bacillus* species in the secondary metabolic gene clusters. Eight secondary metabolic gene clusters responsible for surfactin, terpenes, bacillaene, fengycin, T3PKS, bacillibactin, and bacilysin are present in all three *Bacillus* species (Figure 9C). Thirteen of the sixteen secondary metabolite gene clusters in SF334 are present in three *B*. *velezensis* strains (Figure 9C). The secondary metabolic gene clusters contained by SF334 are highly similar to FZB42, with only one synthesizing more kijianimicin than FZB42. These results suggest that SF334 may have similar biocontrol functions relative to FZB42 in inhibiting microorganisms, inducing host resistance and promoting plant growth.

## 4. Discussion

In this study, we isolated and identified a strain of *B. velezensis* SF334, which had a good prevention effect on the leaf anthracnose of rubber trees caused by *C*. *siamense* and *C*. *australisinense*. In addition, we explored the antagonistic mechanism of SF344 against *C*. *siamense* and *C*. *australisinense* using microscopic observation, including scanning electron microscopy under the interaction system between biocontrol bacteria and pathogenic fungi. To our knowledge, this is the first report of a *B. velezensis* strain as a potential biocontrol agent of the leaf anthracnose of rubber trees caused by *C*. *siamense* and *C*. *australisinense*, which laid a good foundation for the green control of the leaf anthracnose of rubber trees.

So far, there are few cases of the biological control of anthracnose in rubber trees. However, some *B. velezensis* strains, for instance, *B. velezensis* CE100 [29], PW192 [30], and HN-2 [31], have been reported to have a high potential as biocontrol agents for anthracnose diseases caused by *C*. *gloeosporioides*. In this study, we tested the control effect of SF334 on the anthracnose of rubber trees by inoculating *C*. *siamense* and *C*. *australisinense* with the Pre and Tre strategies, respectively, and found that efficacy could reach more than 70% using the Pre strategy, which was significantly higher than that exhibited by the Tre strategy. We speculated that the reason for the low efficacy in the Tre strategy was due to our inoculation method, in which the inoculated mass may have provided a higher fungal initial source, resulting in less effectiveness in the subsequent spraying or application of SF334. In the Pre strategy, SF334, which first existed on the leaves, killed part of the initial sources of fungi and simultaneously blocked or delayed the infection of *C*. *siamense* and *C*. *australisinense* on the leaves of rubber trees. Another speculation is that in this strategy, SF334 might trigger induced systemic resistance in the rubber trees, as some cyclic lipopeptides (i.e., surfactin, fengycin, bacillomycin-D) and volatile organic compounds (VOCs) from two model strains of *B. velezensis* FZB42 and SQR9 have been reported to induce resistance in plants [28]. It is possible that surfactin prevents fungal attachment as a surfactant agent. Therefore, the key to the prevention and control of leaf anthracnose lies in the early stage of the disease, when the initial number of fungi is not large. The preventive management of *B. velezensis* as a biopesticide might have an ideal effect.

In this study, we observed an interesting appearance that SF334 can cause the lysis of hyphae when interacting with *C*. *siamense* and *C*. *australisinense*. Observations using a microscope and scanning electron microscopy revealed that SF334 caused the swollen structures of mycelium; in particular, circular expansion at the growth point of mycelium is the most obvious. The swollen mycelium can be dyed blue using Evans blue, indicating the death of the mycelium. Similar results were observed when the lipopeptide bacillomycin D, produced by *B. velezensis* FZB42 and HN-2, was shown to interact with the hyphae of *F*. *graminearum* and *C*. *gloeosporioides* [31,32], indicating that bacillomycin D can cause damage to the cell wall and membranes, resulting in the leakage of the cytoplasm. Conidia could be observed in the control mycelium, but mycelium expansion after SF334 treatment was much larger than that of raw mycelium. We speculated that mycelium expansion, which may occur at the middle or top of the mycelium, with increased frequency at the top, was caused by bacillomycin D generated by SF334, and it is this enlargement of the apex that inhibits conidial formation. Some studies have also found that biosurfactants (i.e., fengycin A and fengycin B) and VOCs (i.e., 5-nonylamine and 3-methylbutanoic acid) produced by *B. velezensis* exhibited antifungal activity against *C*. *gloeosporioides*, the causal agent of the anthracnose disease of fruit trees [29,30]. We found that SF334 secretes cellulase and chitinase, which can degrade the cell walls of fungi and oomycetes. Whether the malformed mycelium of *C*. *siamense* and *C*. *australisinense* treated by SF334 is due to bacillomycin D or other biosurfactants, as well as cellulase or chitinase, remains to be further explored. We speculated that this digestion effect may not be due to the action of pectinase, because in our other *Bacillus velezensis* SF305 (data not shown), it can degrade the mycelia of *Ganoderma pseudoferreum*, causing the red root disease of rubber trees. When the interaction of SF305 and *G. pseudoferreum* arrived at 12 h, the morphology of the fungus could not be observed under a scanning electron microscope. However, after the interaction between SF334 and the mycelia of *C*. *australisinense* and *C*. *siamense* for 12 h, although the hypha dies, the whole form remains intact.

*B. velezensis* is a versatile plant probiotic Gram-positive bacterium that can produce 13 known bioactive molecules, including cyclic lipopeptides (i.e., surfactin, fengycin, bacillomycin-D, bacillibactin), polyketides (i.e., difficidin, bacillaene, macrolactin, kijanimicin), dipeptide antibiotics (bacilysin), and bacteriocins (i.e., plantazolicin, amylocyclicin); saccharides (butirosin A/B); and a new NRPS (bacillothiazols), as well as VOCs (i.e., acetoin, 2,3-butandiol) [28], which have been reported to exhibit a broad spectrum of antagonistic microbial activity. The antagonism of the model *B*. *velezensis* strain FZB42 against *Rhizoctonia solani* causing the bottom rot disease of lettuce and *F. graminearum*, a plant pathogenic fungus of wheat and barley, has been reported due to the presence of fengycin and bacillomycin-D [32,33]. Bacilysin and difficidin produced by FZB42 were reported to exert a biocontrol effect against bacterial blight and bacterial leaf streak caused by *Xoo* and *Xoc*, respectively [26]. FZB42 was also reported to exert an inhibitory effect against soybean pathogen *Phytophthora sojae* and pear pathogen *Erwinia amylovora* due to difficidin production [27,28]. Our comparative genomic analysis showed that SF334 is more closely related to FZB42 than SQR9, possessing nine secondary metabolites gene clusters with 100% similarity relative to the ones of FZB42. In addition, we found that SF334 can secrete protease and siderophore and produce IAA, which can promote plant growth, and it has antagonistic effects on some productively important plant pathogenic fungi (i.e., *M*. *oryzae*, *B*. *cinerea*, *P*. *capsici*, *F*. *graminearum*, *F*. *oxysporum* f. sp. *spcucumerinum*), suggesting that SF334 is also a multifunctional plant probiotic bacterium.

## 5. Conclusions

In summary, we isolated *B*. *velezensis* SF334 and demonstrated that it is a potential biocontrol agent for the leaf anthracnose of rubber trees. This study not only found a new biocontrol effect of *B*. *velezensis* but also provided ideas for the future green prevention and control of rubber tree diseases in the field.

## Figures and Tables

**Figure 1 jof-10-00158-f001:**
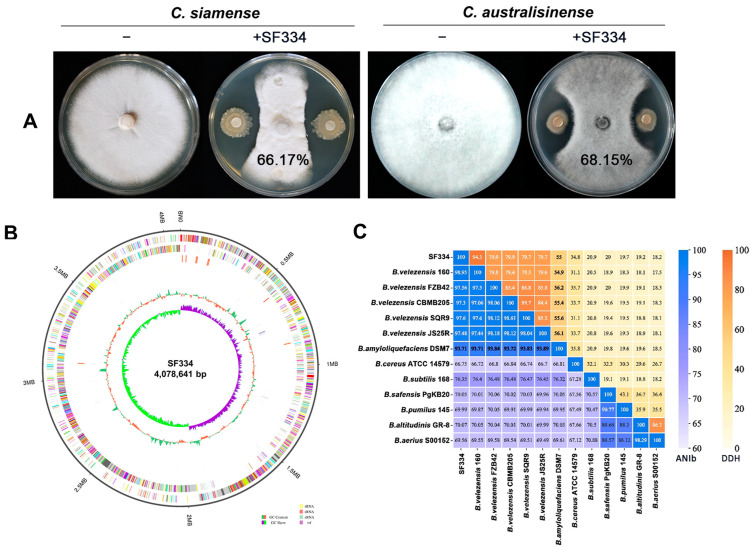
Screening and identification of strain SF334 that exhibits antagonistic activity against *C*. *siamense* and *C*. *australisinense*. (**A**) Antagonistic activity of strain SF334 against *C*. *siamense* and *C*. *australisinense*, which are major pathogens causing leaf anthracnose of rubber trees in the Hainan province of China. The average inhibition rates of SF334 against *C*. *siamense* and *C*. *australisinense* were indicated on the plates. (**B**) The genome features of strain SF334. The outer to inner circles, respectively, indicate genome side (1), forward strands, (2) and reverse strands (3) colored according to the cluster of orthologous group (COG) category, sense strand non-coding RNAs (4), anti-sense strand non-coding RNAs (5), repeat sequence region (6), the GC content (7), and the GC skew in green (+) and purple (−) (8), respectively. (**C**) The ANIb and DDH values of SF334 with 13 *Bacillus* strains including 5 strains of *B*. *velezensis*.

**Figure 2 jof-10-00158-f002:**
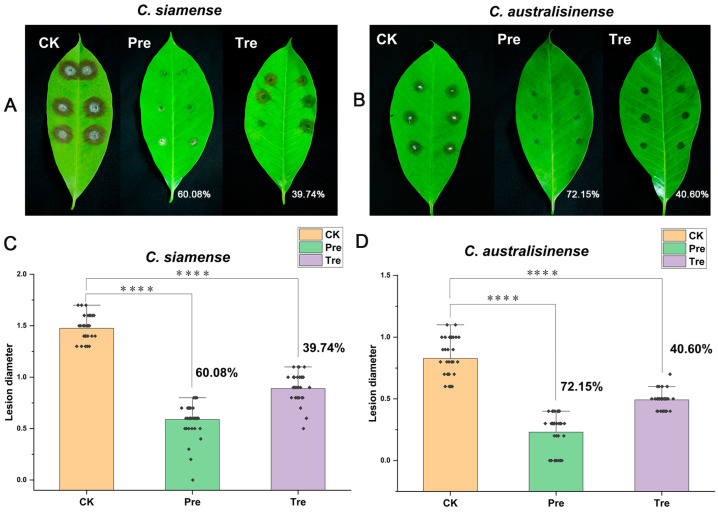
The biocontrol assays of SF334 on Brazilian rubber trees based on detached leaves. The in vitro leaves of Brazilian rubber trees were inoculated with either *C*. *siamense* (**A**) or *C*. *australisinense* (**B**) for 72 h. The biocontrol efficiencies were calculated according to the lesions’ diameters relative to *C*. *siamense* (**C**) or *C*. *australisinense* (**D**). The treatment (Tre) strategy meant that leaves were sprayed with the cell suspensions (CSs) of SF334 (OD_600_ = 1.0) 24 h after inoculation with either *C*. *siamense* or *C*. *australisinense*, and the preventive (Pre) strategy indicated that leaves were sprayed with the CSs of SF334 24 h before inoculation with either *C*. *siamense* or *C*. *australisinense*. The Pre and Tre efficacies are indicated. **** indicates statistically significant differences with *p* < 0.0001.

**Figure 3 jof-10-00158-f003:**
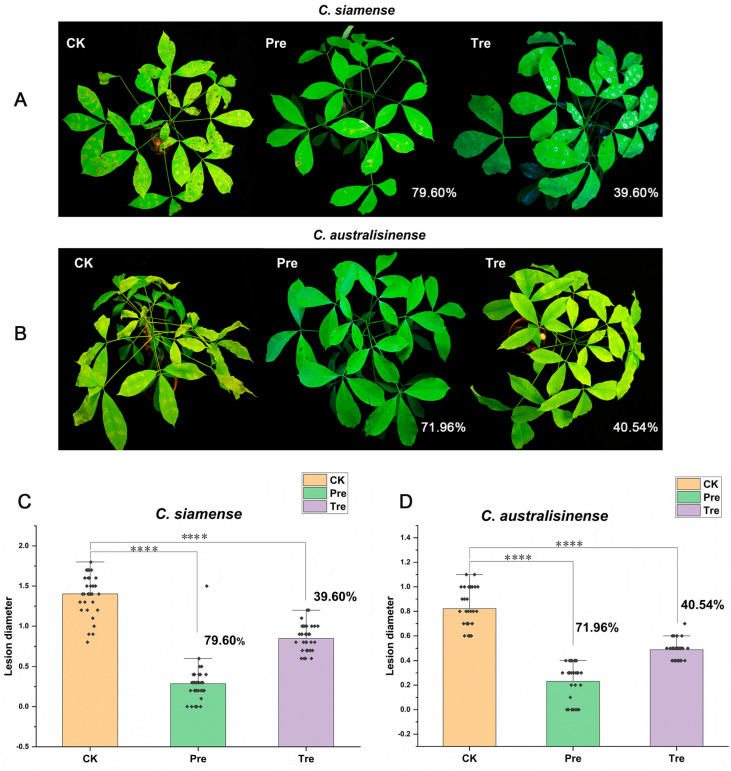
The biocontrol effect of SF334 on Brazilian rubber trees against leaf anthracnose caused by *C*. *siamense* and *C*. *australisinense*. (**A**) Leaf anthracnose was caused by *C*. *siamense.* (**B**) Leaf anthracnose was caused by *C*. *australisinense.* (**C**) The biocontrol efficiencies of SF334 were calculated according to the lesions’ diameters relative to *C*. *siamense*. (**D**) The biocontrol efficiencies of SF334 were calculated according to the lesions’ diameters relative to *C*. *australisinense*. The treatment (Tre) strategy meant that leaves were sprayed with the cell suspensions (CSs) of SF334 (OD_600_ = 1.0) 24 h after inoculation with either *C*. *siamense* or *C*. *australisinense*, and the preventive (Pre) strategy indicated that leaves were sprayed with the CSs of SF334 24 h before inoculation with either *C*. *siamense* or *C*. *australisinense*. **** indicates statistically significant differences with *p* < 0.0001.

**Figure 4 jof-10-00158-f004:**
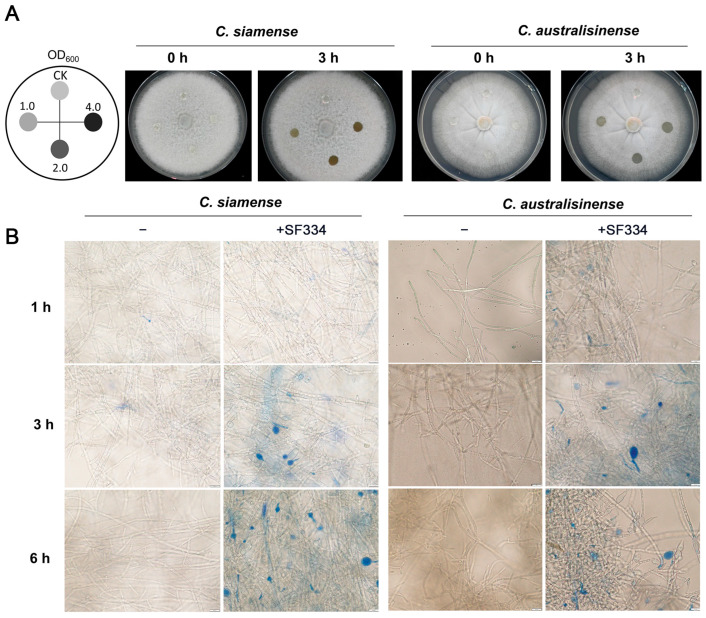
Analysis of the antagonistic mechanism of *B. velezensis* SF334 against *C*. *siamense* and *C*. *australisinense*. (**A**) Observation of the hyphal lysis of *C*. *siamense* and *C*. *australisinense* when interacting with the CSs of *B. velezensis* SF334 on PDA plates. The mycelia of *C*. *siamense* and *C*. *australisinense* were inoculated with the indicated bacterial concentration for 0 h and 3 h. (**B**) Observation of mycelium morphology of *C*. *siamense* and *C*. *australisinense* using an optical microscope when interacting with the CSs of *B. velezensis* SF334 in PDB medium with or without 0.05% Evans blue staining. Scale bar, 10 μm.

**Figure 5 jof-10-00158-f005:**
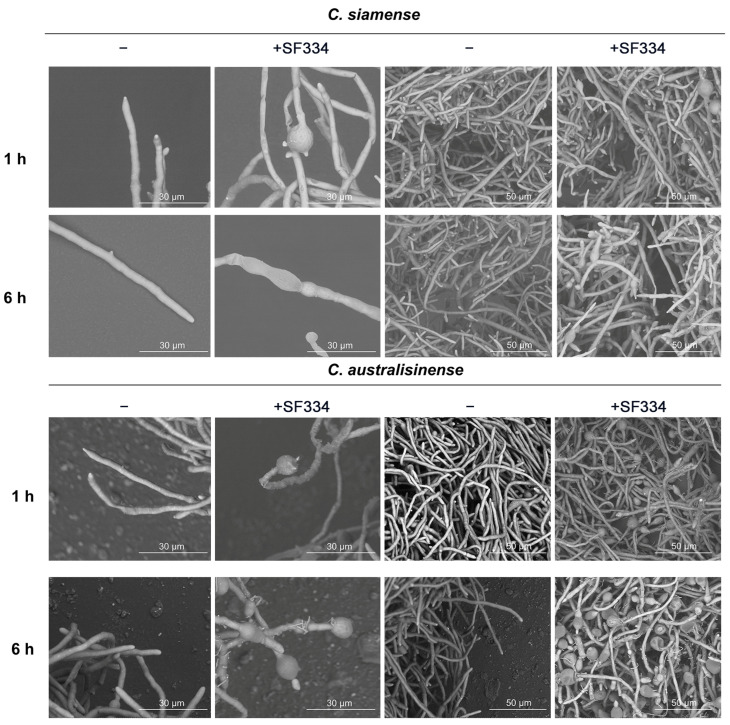
Observation of mycelium morphology of *C*. *siamense* and *C*. *australisinense* using a scanning electron microscope when interacting with the CSs of *B. velezensis* SF334 for 1 h and 6 h in PDB medium. Scale bars of 10 μm and 50 μm are indicated.

**Figure 6 jof-10-00158-f006:**
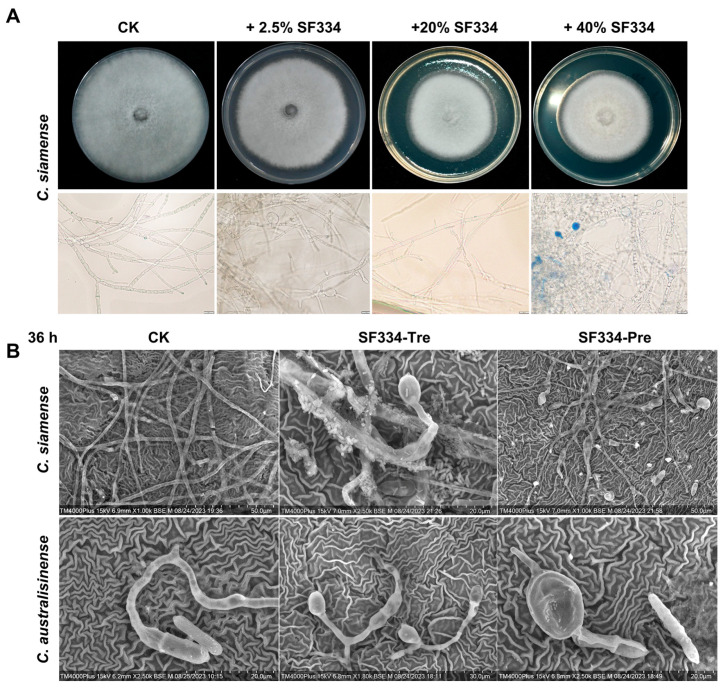
The mycelial expansion phenotypes were observed when using the cell-free supernatants (CFSs) of SF334 to interact with *C. siamense* and when spraying the CFSsof SF334 on the live leaves of Brazilian rubber trees. (**A**) The inhibitory effect against *C*. *siamense* of the cell-free supernatants of SF334. The mycelium of *C*. *siamense* on PDA plates (upper) and in PDB medium (lower) with 0.05% Evans blue staining observed using an optical microscope. The PDA plates or PDB medium were mixed with the CFSs of SF334 via the indicated volume ratios. (**B**) Observation of mycelium morphology of *C*. *siamense* and *C*. *australisinense* when interacting with the CSs of *B. velezensis* SF334 for 36 h on live leaves of Brazilian rubber trees using a scanning electron microscope. Scale bars of 20 μm, 30 μm, and 50 μm were indicated.

**Figure 7 jof-10-00158-f007:**
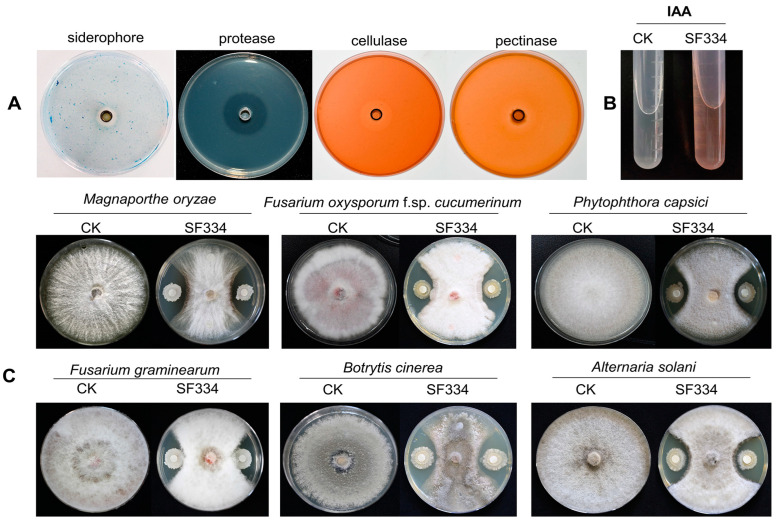
Analyses of plant probiotic characteristics and antifungal activity of *B. velezensis* SF334. (**A**) The plant probiotic characteristics of *B. velezensis* SF334, including siderophore production and protease, cellulase, and pectinase activities, were examined on agar plates. (**B**) The auxin of indole-3-acetic acid (IAA) production of *B. velezensis* SF334 was tested in YM medium. (**C**) The antifungal activities of *B. velezensis* SF334 against five plant pathogenic fungi were measured on PDA plates. *M*. *oryzae* causing rice blast, *F*. *oxysporum* f. sp. *spcucumerinum* causing the root rot disease of cucumber, *P*. *capsici* causing pepper phytophthora blight, *F*. *graminearum* causing fusarium head blight, *B*. *cinerea* causing the gray mold disease of vegetables, and *A*. *solani* causing early blight of potato were used as the target pathogens.

**Figure 8 jof-10-00158-f008:**
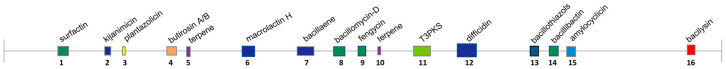
Secondary metabolite biosynthesis gene clusters of *B*. *velezensis* SF334 predicted by AntiSMASH. The active molecules related to the gene cluster (color boxes) and the cluster regions according to Table 2 are indicated above and below the horizontal line, respectively. T3PKS, type III polyketide synthase cluster; NRPS, non-ribosomal peptide synthetase cluster.

**Figure 9 jof-10-00158-f009:**
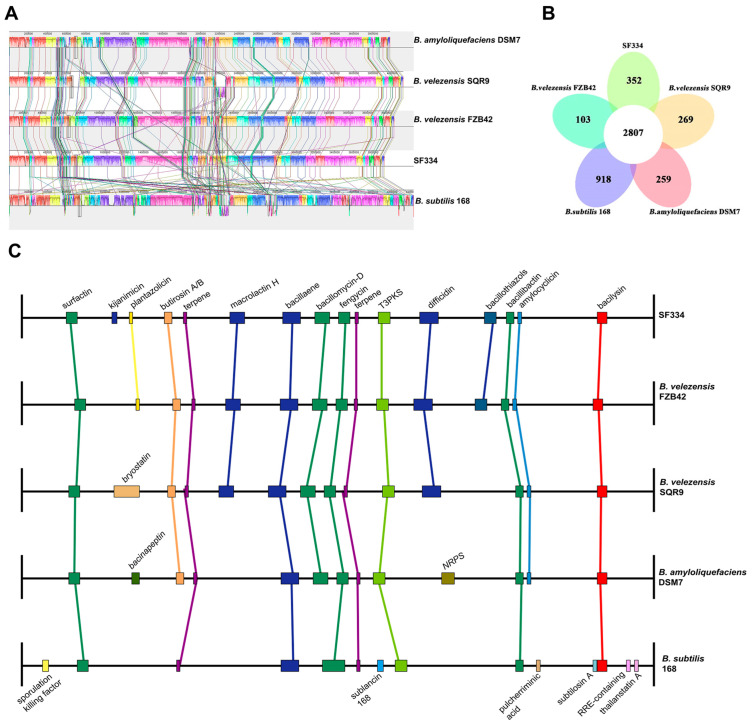
Comparative genomic analysis of *B. velezensis* SF334 with four other related *Bacillus* species. (**A**) Genome-to-genome alignment of *B*. *velezensis* SF334 with *B*. *velezensis* FZB42, *B*. *velezensis* SQR9, *B*. *amyloliquefaciens* DSM7, and *B*. *subtilis* 168. Boxes with the same color indicate the syntenic regions. (**B**) Pan-genomic analysis showing the number of genes of orthologous CDSs that are shared and unique between five strains. (**C**) Comparison of the secondary metabolite biosynthesis gene clusters of *B*. *velezensis* SF334 with *B*. *velezensis* FZB42, *B*. *velezensis* SQR9, *B*. *amyloliquefaciens* DSM7, and *B*. *subtilis* 168. The same gene clusters are indicated by the same boxes and lines. T3PKS, type III polyketide synthase cluster; NRPS, non-ribosomal peptide synthetase cluster.

**Table 1 jof-10-00158-t001:** General features of the *B*. *velezensis* SF334 genome.

General Features	*B*. *velezensis* SF334
Genome size (bp)	4,078,641
GC content (%)	46.5
Coding density (%)	89.33
Protein coding sequences (CDS)	4142
tRNA	86
5s rRNA	9
16s rRNA	9
23s rRNA	9
sRNA	33
Minisatellite DNA	131
Microsatellite DNA	13
Genes assigned to COGs	3022
Genes assigned to GOs	2376
Genes connected to KEGG pathways	2554
Genes assigned to NR	4122
Gene was assigned to Swiss-Prot	3289
Genes assigned to CAZy	103

**Table 2 jof-10-00158-t002:** Secondary metabolite biosynthesis gene clusters of *B*. *velezensis* SF334 predicted by AntiSMASH.

Cluster	Type	Location	Most Similar Known Cluster	Similarity
Region 1	Lipopeptide (NRPS)	308,103–373,510	Surfactin	82%
Region 2	Polyketid (LAP)	588,937–618,053	Kijanimicin	4%
Region 3	Bacteriocin	702,156–725,333	Plantazolicin	91%
Region 4	Saccharide (PKS-like)	937,179–978,423	Butirosin A/B	7%
Region 5	Terpene	1,060,445–1,081,185	Unknown	ND
Region 6	Polyketid (NRPS)	1,453,684–1,541,890	Macrolactin H	100%
Region 7	Polyketid (NRPS/PKS)	1,763,248–1,873,368	Bacillaene	100%
Region 8	Lipopeptide (NRPS/PKS)	1,951,281–1,995,921	Bacillomycin-D	100%
Region 9	Lipopeptide (NRPS)	2,004,711–2,054,255	Fengycin	100%
Region 10	Terpene	2,094,862–2,116,745	Unknown	ND
Region 11	T3PKS	2,226,268–2,267,368	Unknown	ND
Region 12	Polyketid (NRPS)	2,425,972–2,532,138	Difficidin	100%
Region 13	NRPS	3,021,305–3,071,546	Bacillothiazols	100%
Region 14	Lipopeptide (NRPS)	3,172,133–3,223,925	Bacillibactin	100%
Region 15	Bacteriocin	3,215,390–3,219,562	Amylocyclicin	100%
Region 16	Dipeptide	3,730,464–3,771,882	Bacilysin	100%

NRPS = nonribosomal peptide synthetases; PKS = polyketide synthases; LAP = linear azol(in)e-containing peptides; T3PKS = Type III PKS. ND indicates that a similar gene cluster was not detected in the antiSMASH database.

## Data Availability

The datasets presented in the study can be found online: https://www.ncbi.nlm.nih.gov/bioproject/PRJNA970529. All other data are provided in this article’s Results section and Appendix A.

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
