# Peer review of "A New Biocontrol Agent Bacillus velezensis SF334 against Rubber Tree Fungal Leaf Anthracnose and Its Genome Analysis of Versatile Plant Probiotic Traits"

_jof, 2024, doi:10.3390/jof10020158_

Round 1
Reviewer 1 Report
The manuscript by Wang et al. describes the isolation of the Bacillus velezensis SF strain, with antagonistic activity against C. australisinense and C. siamense, causing foliar anthracnose in rubber trees in China. SF cell supernatants showed significant preventive efficacy (% against C. siamense and % against C. australisinense). SF induces mycelial lysis, highlighting plant growth-promoting traits. Genomic analysis reveals similarity with other B. velezensis strains. SF344 causes hyphal lysis, possibly due to biosurfactants such as bacillomycin D. The efficacy of SF334 was more significant in the preventive strategy. B. velezensis SF334 is suggested to be a promising biocontrol agent for foliar anthracnose in rubber trees. These results are relevant and should be published in the Journal of Fungi. However, I consider it essential that authors take into account the following recommendations to improve their manuscript:
-Perform additional experiments to confirm the possible induction of systemic resistance in rubber trees treated with B. velezensis SF334. This could include gene expression analysis of genes related to resistance and plant immune response.
-Perform additional experiments to understand the mechanisms responsible for hyphal lysis further. Specific assays could be carried out to identify the contribution of individual components, such as bacillomycin D, biosurfactants, or other substances secreted by B. velezensis SF334.
-Perform long-term experiments to evaluate the persistence and duration of protection provided by B. velezensis SF334. This could involve continuous monitoring of the incidence of foliar anthracnose in treated rubber trees over several growing seasons.
-Experiment with different application strategies, concentrations, and frequencies to determine the most effective use of B. velezensis SF334 as a biocontrol agent. This could include optimizing the amount of bacteria applied and the ideal timing for application.
-Perform trials to evaluate the potential toxicity of B. velezensis SF334 to non-target organisms and its impact on the environment. This is essential to ensure this biocontrol agent's environmental safety and practical applicability.
-Investigate the interactions between B. velezensis SF334 and the soil microbiota and other beneficial or pathogenic microorganisms species. Understand how these interactions can affect the long-term effectiveness of biocontrol.
-Conduct large-scale field trials in different locations and climatic conditions to evaluate the effectiveness and consistency of B. velezensis SF334 as a biocontrol agent in natural environments.
-Investigate the genetic variability within the B. velezensis SF334 strain and how it can influence its biocontrol capacity. This could include genomic sequencing individual strains' responses to varying environmental conditions.
-Evaluate the compatibility of B. velezensis SF334 with other control methods used in managing foliar anthracnose, such as chemical fungicides. Determine if there is synergy or antagonism between these methods.
Please correct the grammar in some sections of the manuscript and the quality of the figures due their resolution is low.
Author Response
Reviewer 1
The manuscript by Wang et al. describes the isolation of the Bacillus velezensis SF334 strain, with antagonistic activity against C. australisinense and C. siamense, causing foliar anthracnose in rubber trees in China. SF334 cell supernatants showed significant preventive efficacy (% against C. siamense and % against C. australisinense). SF334 induces mycelial lysis, highlighting plant growth-promoting traits. Genomic analysis reveals similarity with other B. velezensis strains. SF344 causes hyphal lysis, possibly due to biosurfactants such as bacillomycin D. The efficacy of SF334 was more significant in the preventive strategy. B. velezensis SF334 is suggested to be a promising biocontrol agent for foliar anthracnose in rubber trees. These results are relevant and should be published in the Journal of Fungi. However, I consider it essential that authors take into account the following recommendations to improve their manuscript.
Response:The reviewer has given us very valuable suggestions, some of which we are currently implementing, and others also have pointed out the direction for our future researches. We really appreciate.
- Perform additional experiments to confirm the possible induction of systemic resistance in rubber trees treated with B. velezensis SF334. This could include gene expression analysis of genes related to resistance and plant immune response. Response 1: We are conducting this part of the experiment on rubber trees, and the corresponding results will be published separately.
- Perform additional experiments to understand the mechanisms responsible for hyphal lysis further. Specific assays could be carried out to identify the contribution of individual components, such as bacillomycin D, biosurfactants, or other substances secreted by B. velezensis SF334. Response 2: We are in the process of this part of the study. By comparing the transcriptome, we analyzed the similarities and differences of the differentially expressed genes of C. australisinense and C. siamense treated by SF334 respectively. However, we used a mixture of the cell-free supernatants. We have not yet obtained a precise evidence on bacillomycin D, which is a single active compound exhibiting antagonistic activity against C. australisinense and C. siamense.
- Perform long-term experiments to evaluate the persistence and duration of protection provided by B. velezensis SF334. This could involve continuous monitoring of the incidence of foliar anthracnose in treated rubber trees over several growing seasons. Response 3: This is a very good suggestion that we will adopt in the following research.
- Experiment with different application strategies, concentrations, and frequencies to determine the most effective use of B. velezensis SF334 as a biocontrol agent. This could include optimizing the amount of bacteria applied and the ideal timing for application. Response 4: This is a very good suggestion that we will adopt in the following research.
- Perform trials to evaluate the potential toxicity of B. velezensis SF334 to non-target organisms and its impact on the environment. This is essential to ensure this biocontrol agent's environmental safety and practical applicability. Response 5: This is a very good suggestion and one that we will definitely study in the future.
- Investigate the interactions between B. velezensis SF334 and the soil microbiota and other beneficial or pathogenic microorganism’s species. Understand how these interactions can affect the long-term effectiveness of biocontrol. Response 6: This is a very interesting topic and a real challenge to our current capabilities, but it is a good idea and we will seriously consider it.
- Conduct large-scale field trials in different locations and climatic conditions to evaluate the effectiveness and consistency of B. velezensis SF334 as a biocontrol agent in natural environments. Response 7: This is a very good suggestion and one that we will definitely study in the future. We really appreciate.
- Investigate the genetic variability within the B. velezensis SF334 strain and how it can influence its biocontrol capacity. This could include genomic sequencing individual strains' responses to varying environmental conditions. Response 8: This is a good suggestion, and we are studying it accordingly.
- Evaluate the compatibility of B. velezensis SF334 with other control methods used in managing foliar anthracnose, such as chemical fungicides. Determine if there is synergy or antagonism between these methods. Response 9: What a coincidence! We used a similar strategy for the next spring test of plant protection and flight prevention.
- Please correct the grammar in some sections of the manuscript and the quality of the figures due their resolution is low. Response 10: We have used Grammarly software to check and modify the grammar of the full text. Our pictures are all HD pictures, which meet the requirements of the magazine for pictures. Maybe the definition becomes smaller when inserted into the word manuscript. We will upload the pictures online and submit them to the magazine.

Reviewer 2 Report
The article “A New Biocontrol Agent Bacillus velezensis SF334 Against Fungal Leaf Anthracnose of Rubber Tree and Its Genome Analysis for Versatile Plant Probiotic Traits” is devoted to the important and acute theme of protection of plants against anthracnose with beneficial microbes. Authors provide and analyze a lot of information on this theme, but minor editions should be made.
I have some comments:
1) Title: I recommend to focus on genome analysis (may be “Genome Analysis of A New Biocontrol Agent Bacillus velezensis SF334 Against Fungal Leaf Anthracnose of Rubber Tree”)
2) Abstract: Grammatics must be checked carefully.
3) Bacillus velezensis SF334 produces cellulase, not cellulose, doesn't it? (abstract, fig. 7 etc)
4) Lines 356-357: Fungal cell walls don't consist of pectin and cellulose. These compounds form cell walls of plants (and cellulose characteristic of oomycetes). Please, put out a hypothesis about the beneficial role of plant cell walls degrading enzymes in plant-microbe interactions.
5) In my opinion, “deformed and bulbous” mycelium tips can be conidia of Colletotrichum (for example, fig 4 in https://www.nature.com/articles/s41598-020-66761-9). Thus, B. velesensis SF334 can stimulate conidia formation. It should be discussed.
6) Mat&met: Compositions of nutrient agar (NA) and nutrient broth (NB) should be given (or manufacturer and catalog#). Please, insert Lysogeny broth instead of Luria-Bertani medium.
1) Title: I recommend to focus on genome analysis (may be “Genome Analysis of A New Biocontrol Agent Bacillus velezensis SF334 Against Fungal Leaf Anthracnose of Rubber Tree”)
2) Abstract: Grammatics must be checked carefully.
3) Bacillus velezensis SF334 produces cellulase, not cellulose, doesn't it? (abstract, fig. 7 etc)
4) Lines 356-357: Fungal cell walls don't consist of pectin and cellulose. These compounds form cell walls of plants (and cellulose characteristic of oomycetes). Please, put out a hypothesis about the beneficial role of plant cell walls degrading enzymes in plant-microbe interactions.
5) In my opinion, “deformed and bulbous” mycelium tips can be conidia of Colletotrichum (for example, fig 4 in https://www.nature.com/articles/s41598-020-66761-9). Thus, B. velesensis SF334 can stimulate conidia formation. It should be discussed.
6) Mat&met: Compositions of nutrient agar (NA) and nutrient broth (NB) should be given (or manufacturer and catalog#). Please, insert Lysogeny broth instead of Luria-Bertani medium.
Author Response
Reviewer 2
The article “A New Biocontrol Agent Bacillus velezensis SF334 Against Fungal Leaf Anthracnose of Rubber Tree and Its Genome Analysis for Versatile Plant Probiotic Traits” is devoted to the important and acute theme of protection of plants against anthracnose with beneficial microbes. Authors provide and analyze a lot of information on this theme, but minor editions should be made.
Response:According to the reviewer's opinions, we have made modifications, but we maintain our original ideas on the title of this manuscript. The corresponding speculation and discussion are added in the discussion section.
I have some comments:
- Title: I recommend to focus on genome analysis (may be “Genome Analysis of A New Biocontrol Agent Bacillus velezensis SF334 Against Fungal Leaf Anthracnose of Rubber Tree”). Response 1:We maintained the original idea and did not revise the title, because only half of the paper was genome analysis, and the other half was analysis of biocontrol efficiency and antagonistic mechanism.
- Abstract: Grammatics must be checked carefully. Response 2: We have used Grammarly software to check and modify the grammar of the full text.
- Bacillus velezensis SF334 produces cellulase, not cellulose, doesn't it? (abstract, fig. 7 etc). Response 3: We really made mistake, and we corrected all the mistakes in the manuscript including Fig. 7.
- Lines 356-357: Fungal cell walls don't consist of pectin and cellulose. These compounds form cell walls of plants (and cellulose characteristic of oomycetes). Please, put out a hypothesis about the beneficial role of plant cell walls degrading enzymes in plant-microbe interactions. Response 4: We corrected this error to “SF334 also could secrete cellulase, and pectinase (Figure 7A), indicating that SF334 can degrade the components of cell walls from fungi and oomycetes.” We added the hypothesis of this part to the discussion section. We speculated that this digestion effect may not be due to the action of pectinase, because in our other Bacillus velezensis SF305 (will be published in other journal), it can degrade the mycelia of Ganoderma pseudoferreum causing the red root disease of rubber tree. When the interaction arrived 12 hr, the morphology of the fungus cannot be seen under scanning electron microscope. However, after the interaction between SF334 and the mycelia of C. australisinense and C. siamense for 12 hr, although the hypha dies, the whole form remains intact.
- In my opinion, “deformed and bulbous” mycelium tips can be conidia of Colletotrichum (for example, fig 4 inhttps://www.nature.com/articles/s41598-020-66761-9). Thus, B. velesensis SF334 can stimulate conidia formation. It should be discussed. Response 5: Conidia could be observed in the control mycelium, but mycelium expansion after SF334 treatment was much larger than that of raw mycelium. We speculated that mycelium expansion was caused by bacillomycin D, an active compound generated by SF334, which may be in the middle or top of mycelium. But there's more frequency at the top. It is this enlargement of the apex that inhibits conidial formation. We also consulted the relevant literatures, and for example, Fig. 3 in the paper of Bacillomycin D Produced by Bacillus amyloliquefaciens Is Involved in the Antagonistic Interaction with the Plant-Pathogenic Fungus Fusarium graminearum and Fig. 4 in the paper of Antifungal mechanism of bacillomycin D from Bacillus velezensis HN-2 against Colletotrichum gloeosporioides Penz. are very similar to what we saw in our article. The corresponding speculation and discussion are added in the discussion section.
- Mat&met: Compositions of nutrient agar (NA) and nutrient broth (NB) should be given (or manufacturer and catalog#). Please, insert Lysogeny broth instead of Luria-Bertani medium. Response 6: We supplemented the NA and NB compositions in Mat&met, and corrected Luria-Bertani medium to Lysogeny broth.

Reviewer 3 Report
The manuscript titled “A New Biocontrol Agent Bacillus velezensis SF334 Against Fungal Leaf Anthracnose of Rubber Tree and Its Genome Analysis for Versatile Plant Probiotic Traits” by Wang et al. deals with the study of a bacterial strain capable to inhibit plant-pathogenic fungi, both in vitro and in planta. The authors also provide the genomic characterization of the bacterial strain as well as they analyze the physiological changes of the target Colletotrichum species that are antagonized. Overall, the manuscript shows a complete description of the biocontrol potential of this Bacillus strain, and I think it can be published in JoF after checking the following issues:
- L 69. Actinobacteria are also bacteria, there is no need to split both groups. In case you want to do so, you should mention “concretely…” or some similar word to talk about Actinobacteria and not be confusing or incorrect.
- L 75. Which species?
- L 85. ‘We’, upper case.
- L 90. ‘It’, upper case.
- L 100 and 105. The symbol for degrees is not correct here.
- L 101. sp.
- L 118. ‘sequence’, in singular?
- L 121. ‘16S rDNA’
- Figure S2. Scale bar is missing
- L 125. The ANI method is missing. Was it ‘ANIb’, which is based on blast searches?
- L 137-138. Even these methods were made by the company, it would be better if you detail such methods here.
- L 141. Results for phages are not shown.
- L 152. Please, italicize the species name.
- L 156. BPGA is a tool or software, not a database. Same for antiSMASH (L158).
- How did you made the CAZy searches? Did you use local blastp searches against the database? Did you use dbCAN2? Please, state it.
- Section 2.8. Hydrolysis halo (line 211) can not refer to all these tests (eg. Phosphate solubilization, siderophores production). Please, provide details on what do you consider as a positive or as a negative result. Is it needed to use some reactive with some of these tests to reveal positive results?
- L 216. How do you relate color with IAA? Might it be related with indole structures, but not so specific to detect IAA? Please, provide more details on this method. The,n on line 355-356 you should not use ‘quantitative’ to talk about IAA if you do not know if other molecules might be also detected with this method.
- L 222. Please, provide a brief description of this method.
- L 228. it would be nice to know the species identification of these strains, and ot know which gave positive results.
- L 231. How do you calculate this ratio? If it is dependent on the ratio of the size of the non-inhibited vs inhibited area, I would say that it should not be correct, since it would depend on the bacterial colony location, and its size. Also, it seems that C. siamense inhibition is greater than the other (Fig 1a), but the inhibition ratio is higher for C. australisiense. I would say there is no need to quantify such inhibition, which just provides confusion. Same for lines 361-368.
- L 238-244. These analyses are meaningless if you do not state the type strain of each species. All these B. velezensis strains might not be such species. It should be needed to compare your genome just against type strains.
- Taxonomy of the strain. I uploaded the genome to the TYGS server and I found that 3 species share > 70 dDDH with the authors’ strain, as well as cluster in the phylogenomic tree with them. It is possible that some of these species are wrongly identified. Please, provide more details and explanation on this issue.
- Figure 1B. This figure is not so essential and lacks quality (maybe just because of the file format conversion in submission procedure). I suggest moving this to the supplementary material. And please, see my above mentioned comment on ANI and dDDH analyses, which might imply that Figure 1C is no longer needed, and if modified, it might also fit in the supplementary material.
- Figure 3. Panel ‘A’ lacks its citation in the caption.
- Figure 4. Scale bar is really hard to see.
- L 350-351. PGPR tests are not related with biocontrol, but with plant growth promotion. I would modify slightly this sentence. Also, the use of PGPR should be changed for PGP along the entire manuscript, since that ‘R’ refers to rhizobacteria, not with PGP mechanisms.
- L 356 – 358. Celluloses can also break plant tissues, which might be beneficial if allows ‘good’ bacteria entering into the plant, or detrimental.
- Figure 5. Please, increase text size
- Table 1. This table might fit better in the supplementary material.
- L450. ‘amylocyclicinand’ -> ‘amylocyclicin and’
- L 453-455. I have run antiSMASH with your genome and I found a BGC equal to the one producing bacillothiazoles and another one almost equal to the BGC producing plantazolicin. You might cite this on these lines. However, I did not find neither amylocyclicin nor bacillomycin-D producing BGCs. Please, check this. In these analyses, 7 BGCs reached 100% of similarity and 2 BGCs are a bove 80%. You can use the following link for a few days:
- L 464. Table 2 instead of S2?
- L 439. Please, use CAZy instead of CAzy here and along the entire manuscript.
- Figure S6. There are many annotations that should not be correct, such as many related with human or animal functioning (eg nervous or digestive systems, cancers, etc).
- L 539. Is it possible that surfactin prevent fungal attachment as a surfactant agent?
- L 569. Bacilysin should be in upper case.
Author Response
Reviewer 3
The manuscript titled “A New Biocontrol Agent Bacillus velezensis SF334 Against Fungal Leaf Anthracnose of Rubber Tree and Its Genome Analysis for Versatile Plant Probiotic Traits” by Wang et al. deals with the study of a bacterial strain capable to inhibit plant-pathogenic fungi, both in vitro and in planta. The authors also provide the genomic characterization of the bacterial strain as well as they analyze the physiological changes of the target Colletotrichum species that are antagonized. Overall, the manuscript shows a complete description of the biocontrol potential of this Bacillus strain, and I think it can be published in JoF after checking the following issues:
Response: We did make a lot of grammar and spelling mistakes, which we corrected in the revised manuscript. The reviewer gave many valuable suggestions, and we made corresponding modifications according to these suggestions, making our revision draft better.
- L 69. Actinobacteria are also bacteria, there is no need to split both groups. In case you want to do so, you should mention “concretely…” or some similar word to talk about Actinobacteria and not be confusing or incorrect. Response 1: We really made mistake, then we deleted the word of “Actinobacteria” in the revised manuscript.
- L 75. Which species? Response 2: We added the corresponding content “An endophytic fungal strain Epicoccum dendrobii SMEL1 from a young heathy branch of Chinese fir (Cunninghamia lanceolata) was found to have a highly antagonistic effect on C. gloeosporioides”.
- L 85. ‘We’, upper case. Response 3: We have corrected this mistake in the revised manuscript.
- L 90. ‘It’, upper case. Response 4: We have corrected this mistake in the revised manuscript.
- L 100 and 105. The symbol for degrees is not correct here. Response 5: We really made mistake, then we have corrected this mistake in the revised manuscript.
- L 101. sp. Response 6: We have corrected this mistake in the revised manuscript.
- L 118. ‘sequence’, in singular? Response 7: We have corrected this mistake in the revised manuscript.
- L 121. ‘16S rDNA’ Response 8: We have corrected all mistakes in the revised manuscript.
- Figure S2. Scale bar is missing Response 9: We have corrected all mistakes in the revised manuscript.
- L 125. The ANI method is missing. Was it ‘ANIb’, which is based on blast searches? Response 10: This is the ‘ANIb’, we corrected this error in the revised manuscript and Fig.1.
- L 137-138. Even these methods were made by the company, it would be better if you detail such methods here. Response 11: We added the company name Personalbio (Shanghai, China) in the revised manuscript.
- L 141. Results for phages are not shown. Response 12: We deleted that part of phages in the revised manuscript.
- L 152. Please, italicize the species name. Response 13: We corrected this error in the revised manuscript.
- L 156. BPGA is a tool or software, not a database. Same for antiSMASH (L158). Response 14: We corrected these errors in the revised manuscript.
- How did you make the CAZy searches? Did you use local blastp searches against the database? Did you use dbCAN2? Please, state it. Response 15: We used the dbCAN2, an online annotation tool in the website http://bcb.unl.edu/dbCAN2/, and we added these information in the revised manuscript.
- Section 2.8. Hydrolysis halo (line 211) can not refer to all these tests (eg. Phosphate solubilization, siderophores production). Please, provide details on what do you consider as a positive or as a negative result. Is it needed to use some reactive with some of these tests to reveal positive results? Response 16: We added the details in the revised manuscript.
- L 216. How do you relate color with IAA? Might it be related with indole structures, but not so specific to detect IAA? Please, provide more details on this method. Then on line 355-356 you should not use ‘quantitative’ to talk about IAA if you do not know if other molecules might be also detected with this method. Response 17: We added the details on the IAA measure. If the solution appears pink, the solution contains indolic compounds. The IAA concentration was calculated according to the standard curve formula y=0.0157x+0.1137 obtained using ten standard concentration of IAA, where Y is the measured UV absorbance in OD530, X is the IAA concentration value. This is a quantitative measurement method, so we have modified the results and added the quantitative and qualitative statements.
- L 222. Please, provide a brief description of this method. Response 18: We gave a reference, this is a paper that we just published last year, and it goes into great details.
- L 228. it would be nice to know the species identification of these strains, and ot know which gave positive results. Response 19: For brevity of the article, we have omitted the details of these 69 strains. The information about these strains will be revealed gradually in our future articles.
- L 231.How do you calculate this ratio? If it is dependent on the ratio of the size of the non-inhibited vs inhibited area, I would say that it should not be correct, since it would depend on the bacterial colony location, and its size. Also, it seems that C. siamense inhibition is greater than the other (Fig 1a), but the inhibition ratio is higher for C. australisiense. I would say there is no need to quantify such inhibition, which just provides confusion. Same for lines 361-368. Response 20: The reviewer's point of view is worth considering, especially with regard to the bacterial colony location, and its size. We avoided this error as much as possible in the experiments and discarded plates with large errors. We obtained this inhibition ratio by means of average values, which we think should be calculated and added in the article because we obtained SF334 strains from 223 candidate strains through this method.
- L 238-244. These analyses are meaningless if you do not state the type strain of each species. All these B. velezensis strains might not be such species. It should be needed to compare your genome just against type strains. Response 21: This analysis is very valid and accurate, and it is our experience from many failed identifications. In the identification of 223 candidate strains, we made a number of errors, because only 16s RNA homology analysis led to the identification of some B. velezensis as B. amyloliquefaciens, but ANI and DDH analyses were able to correct this error. We used the type strain FZB42 of B. velezensis and other B. velezensis strains in this analysis, Similar results were obtained that SF334 belonging to B. velezensis.
- Taxonomy of the strain. I uploaded the genome to the TYGS server and I found that 3 species share > 70 dDDH with the authors’ strain, as well as cluster in the phylogenomic tree with them. It is possible that some of these species are wrongly identified. Please, provide more details and explanation on this issue. Response 22: We submitted SF334 to the TYGS server and found that SF334 belonged to B. velezensis. Based on dDDH analysis, the reviewer should find the dDDH values of the three strains, including B. amyloliquefaciens plantarum FZB4, B. methylotrophicus KACC 13105, and B. velezensis NRRL B-41580, were all greater than 70, which was due to the small genomic differences among the three strains. According to Dunlap et al. (2016) B. amyloliquefaciens and B. methylotrophicus are later heterotypic synonyms of B. velezensis based on phylogenomics. See the reference: Dunlap CA, Kim SJ, Kwon SW, Rooney AP. Bacillus velezensis is not a later heterotypic synonym of Bacillus amyloliquefaciens; Bacillus methylotrophicus, Bacillus amyloliquefaciens subsp. plantarum and 'Bacillus oryzicola' are later heterotypic synonyms of Bacillus velezensis based on phylogenomics. Int J Syst Evol Microbiol. 2016 Mar;66(3):1212-1217.
- Figure 1B. This figure is not so essential and lacks quality (maybe just because of the file format conversion in submission procedure). I suggest moving this to the supplementary material. And please, see my above mentioned comment on ANI and dDDH analyses, which might imply that Figure 1C is no longer needed, and if modified, it might also fit in the supplementary material. Response 23: The Fig.1B and Fig. 1C are so important that we stick with them. We provided the image with high definition and uploaded it to the magazine's website.
- Figure 3. Panel ‘A’ lacks its citation in the caption. Response 24: We added the citation of Fig. 3A, 3B, 3C and 3D in the in the revised manuscript.
- Figure 4. Scale bar is really hard to see. Response 25: We provided the image of Fig. 4 in high definition and uploaded it to the magazine's website.
- L 350-351. PGPR tests are not related with biocontrol, but with plant growth promotion. I would modify slightly this sentence. Also, the use of PGPR should be changed for PGP along the entire manuscript, since that ‘R’ refers to rhizobacteria, not with PGP mechanisms. Response 26: We corrected the “PGPR” to “plant probiotic” along the entire manuscript.
- L 356 – 358. Celluloses can also break plant tissues, which might be beneficial if allows ‘good’ bacteria entering into the plant, or detrimental. Response 27: We added this information in the corresponding Result 3.4 section.
- Figure 5. Please, increase text size. Response 28: We provided the image of Fig. 5 in high definition and uploaded it to the magazine's website.
- Table 1. This table might fit better in the supplementary material. Response 29: This Table is so important that we stick with it as Table 1.
- ‘amylocyclicinand’ -> ‘amylocyclicin and’ Response 30: We really made mistake, then we have corrected this mistake in the revised manuscript.
- L 453-455. I have run antiSMASH with your genome and I found a BGC equal to the one producing bacillothiazoles and another one almost equal to the BGC producing plantazolicin. You might cite this on these lines. However, I did not find neither amylocyclicin nor bacillomycin-D-producing BGCs. Please, check this. In these analyses, 7 BGCs reached 100% of similarity and 2 BGCs are a bove 80%. You can use the following link for a few days: https://antismash.secondarymetabolites.org/upload/bacteria-44907c48-3379-42f6-9fa4-603519c1ddd8/index.html#. Response 31: From this question, we can see that the reviewer is a very rigorous scientific researcher. We also used the same method with the reviewer at the beginning, but did not find neither amylocyclicin nor bacillomycin-D-producing BGCs. However, our reading of the related literature (see below) showed that B. velezensis FZB42 contains amylocyclicin and bacillomycin-D-producing BGCs. Therefore, we conducted further analysis. In Region 1.8 gene cluster, MIBIG comparison tool showed that bacillomycin-D-producing BGC was located in front of fengycin-producing BGC, and was not in the same position as fengycin-producing BGC. According to BGC0001090: bacillomycin D biosynthetic gene cluster from B. velezensis FZB42, the position of bacillomycin-D-producing BGC in SF334 strain was found using the SnapGene software. Similarly, the Amylocyclicin gene cluster is at the back end of the 1.13 gene cluster, which is not in the same location as bacillibactin-producing BGC. We also modified the corresponding Fig.8, Fig.9, Table 2 and the text in the revised manuscript.
The related literatures:
Chowdhury S P, Hartmann A, Gao X, Borriss R. 2015. Biocontrol mechanism by root-associated Bacillus amyloliquefaciens FZB42 - a review. Front Microbiol, 6, 780. doi,10.3389/fmicb.2015.00780.
Fan B, Wang C, Song X, Ding X, Wu L, Wu H, Gao X, Borriss R. 2018. Bacillus velezensis FZB42 in 2018: The Gram-Positive Model Strain for Plant Growth Promotion and Biocontrol. Front Microbiol, 9, 2491. doi,10.3389/fmicb.2018.02491.
Rabbee M F, Ali M S, Choi J, Hwang B S, Jeong S C, Baek K H. 2019. Bacillus velezensis: A Valuable Member of Bioactive Molecules within Plant Microbiomes. Molecules, 24. doi,10.3390/molecules24061046.
- L 464. Table 2 instead of S2? Response 32: we have corrected this mistake in the revised manuscript.
- L 439. Please, use CAZy instead of CAzy here and along the entire manuscript. Response 33: We really made mistake, then we have corrected this mistake in the revised manuscript.
- Figure S6. There are many annotations that should not be correct, such as many related with human or animal functioning (eg nervous or digestive systems, cancers, etc). Response 34: We deleted the part relative to nervous or digestive systems, cancers, etc.
- L 539. Is it possible that surfactin prevent fungal attachment as a surfactant agent? Response 35: It is possible, and we added this information in the corresponding discussion section.
- L 569. Bacilysin should be in upper case. Response 36: We corrected this error in the revised manuscript.
